# STOCHASTIC GRADIENT DESCENT WITH BIASED BUT CONSISTENT GRADIENT ESTIMATORS

## ABSTRACT

Stochastic gradient descent (SGD), which dates back to the 1950s, is one of the most popular and effective approaches for performing stochastic optimization. Research on SGD resurged recently in machine learning for optimizing convex loss functions and training nonconvex deep neural networks. The theory assumes that one can easily compute an unbiased gradient estimator, which is usually the case due to the sample average nature of empirical risk minimization. There exist, however, many scenarios (e.g., graphs) where an unbiased estimator may be as expensive to compute as the full gradient because training examples are interconnected. Recently, Chen et al. (2018) proposed using a consistent gradient estimator as an economic alternative. Encouraged by empirical success, we show, in a general setting, that consistent estimators result in the same convergence behavior as do unbiased ones. Our analysis covers strongly convex, convex, and nonconvex objectives. We verify the results with illustrative experiments on synthetic and real-world data. This work opens several new research directions, including the development of more efficient SGD updates with consistent estimators and the design of efficient training algorithms for large-scale graphs.

## 1 INTRODUCTION

Consider the standard setting of supervised learning. There exists a joint probability distribution $P(x, y)$ of data $x$ and associated label $y$ and the task is to train a predictive model, parameterized by $w$, that minimizes the expected loss $\ell$ between the prediction and the ground truth $y$. Let us organize the random variables as $\xi = (x, y)$ and use the notation $\ell(w; \xi)$ for the loss. If $\xi_i = (x_i, y_i)$, $i = 1, \ldots, n$, are iid training examples drawn from $P$, then the objective function is either one of the following well-known forms:

$$\text{expected risk } f(w) = \mathrm{E}[\ell(w; \xi)]; \quad \text{empirical risk } f(w) = \frac{1}{n} \sum_{i=1}^{n} \ell(w; \xi_i). \tag{1}$$

Stochastic gradient descent (SGD), which dates back to the seminal work of Robbins & Monro (1951), has become the de-facto optimization method for solving these problems in machine learning. In SGD, the model parameter is updated until convergence with the rule[1]

$$w_{k+1} = w_k - \gamma_k g_k, \quad k = 1, 2, \ldots, \tag{2}$$

where $\gamma_k$ is a step size and $g_k$ is an unbiased estimator of the gradient $\nabla f(w_k)$. Compared with the full gradient (as is used in deterministic gradient descent), an unbiased estimator involves only one or a few training examples $\xi_i$ and is usually much more efficient to compute.

### 1.1 LIMITATION OF UNBIASED GRADIENT AND REMEDY: CONSISTENT GRADIENT

This scenario, however, does not cover all learning settings. A representative example that leads to costly computation of the unbiased gradient estimator $\nabla \ell(w, \xi_i)$ is graph nodes. Informally speaking, a graph node $\xi_i$ needs to aggregate information from its neighbors. If information is aggregated

---

[1]For introductory purpose we omit the projection operator for constrained problems. All analysis in this work covers projection.

across neighborhoods, $\xi_i$ must request information from its neighbors recursively, which results in inquiring a large portion of the graph. In this case, the sample loss $\ell$ for $\xi_i$ involves not only $\xi_i$, but also all training examples within its multihop neighborhood. The worst case scenario is that computing $\nabla\ell(w, \xi_i)$ costs $O(n)$ (e.g., for a complete graph or small-world graph), as opposed to $O(1)$ in the usual learning setting because only the single example $\xi_i$ is involved.

In a recent work, Chen et al. (2018) proposed a consistent gradient estimator as an economic alternative to an unbiased one for training graph convolutional neural networks, offering substantial evidence of empirical success. A summary of the derivation is presented in Section 2. The subject of this paper is to provide a thorough analysis of the convergence behavior of SGD when $g_k$ in (2) is a consistent estimator of $\nabla f(w_k)$. We show that using this estimator results in the same convergence behavior as does using unbiased ones.

**Definition 1.** An estimator $g^N$ of $h$, where $N$ denotes the sample size, is *consistent* if $g^N$ converges to $h$ in probability: $\mathrm{plim}_{N\to\infty} g^N = h$. That is, for any $\epsilon > 0$, $\lim_{N\to\infty} \Pr(\|g^N - h\| > \epsilon) = 0$.

## 1.2 Distinctions between Unbiasedness and Consistency

It is important to note that unbiased and consistent estimators are not subsuming concepts (one does not imply the other), *even in the limit*. This distinction renders the departure of our convergence results, in the form of probabilistic bounds on the error, from the usual SGD results that bound instead the *expectation* of the error.

In what follows, we present examples to illustrate the distinctions between unbiasedness and consistency. To this end, we introduce *asymptotic unbiasedness*, which captures the idea that the bias of an estimator may vanish in the limit.

**Definition 2.** An estimator $g^N$ of $h$, where $N$ denotes the sample size, is *asymptotically unbiased* if $\mathrm{E}[g^N] \to h$.

**An estimator can be (asymptotically) unbiased but inconsistent.** Consider estimating the mean $h = \mu$ of the normal distribution $N(\mu, \sigma^2)$ by using $N$ independent samples $X_1, \ldots, X_N$. The estimator $g^N = X_1$ (i.e., always use $X_1$ regardless of the sample size $N$) is clearly unbiased because $\mathrm{E}[X_1] = \mu$; but it is inconsistent because the distribution of $X_1$ does not concentrate around $\mu$. Moreover, the estimator is trivially asymptotically unbiased.

**An estimator can be consistent but biased.** Consider estimating the variance $h = \sigma^2$ of the normal distribution $N(\mu, \sigma^2)$ by using $N$ independent samples $X_1, \ldots, X_N$. The estimator $g^N = \sum_{i=1}^{N}(X_i - \overline{X})^2/N$, where $\overline{X} = \sum_{i=1}^{N} X_i/N$, has mean $\sigma^2(N-1)/N$ and variance $2\sigma^4(N-1)/N^2$. Hence, it is consistent owing to a straightforward invocation of the Chebyshev inequality, by noting that the mean approaches $\sigma^2$ and the variance approaches zero. However, the estimator admits a nonzero bias $\sigma^2/N$ for any finite $N$.

**An estimator can be consistent but biased even asymptotically.** In the preceding example, the bias $\sigma^2/N$ approaches zero and hence the estimator is asymptotically unbiased. Other examples exist for the estimator to be biased even asymptotically. Consider estimating the quantity $h = 0$ with an estimator $g^N$ that takes the value $0$ with probability $(N-1)/N$ and the value $N$ with probability $1/N$. Then, the probability that $g^N$ departs from zero approaches zero and hence it is consistent. However, $\mathrm{E}[g^N] = 1$ and thus the bias does not vanish as $N$ increases.

## 1.3 Contributions of This Work

To the best of our knowledge, this is the first work that studies the convergence behavior of SGD with consistent gradient estimators, which result from a real-world graph learning scenario that will be elaborated in the next section. With the emergence of graph deep learning models (Bruna et al., 2014; Defferrard et al., 2016; Li et al., 2016; Kipf & Welling, 2017; Hamilton et al., 2017; Gilmer et al., 2017; Veličković et al., 2018), the scalability bottleneck caused by the expensive computation of the sample gradient becomes a pressing challenge for training (as well as inference) with large graphs. We believe that this work underpins the theoretical foundation of the efficient training of a series of graph neural networks. The theory reassures practitioners of doubts on the convergence of their optimization solvers.

Encouragingly, consistent estimators result in a similar convergence behavior as do unbiased ones. The results obtained here, including the proof strategy, offer convenience for further in-depth analysis under the same problem setting. This work opens the opportunity of improving the analysis, in a manner similar to the proliferation of SGD work, from the angles of relaxing assumptions, refining convergence rates, and designing acceleration techniques.

We again emphasize that unbiasedness and consistency are two separate concepts; neither subsumes the other. One may trace that we intend to write the error bounds for consistent gradient estimators in a manner similar to the expectation bounds in standard SGD results. Such a resemblance (e.g., in convergence rates) consolidates the foundation of stochastic optimization built so far.

## 2 MOTIVATING APPLICATION: REPRESENTATION LEARNING OF GRAPH NODES

For a motivating application, consider the graph convolutional network model, GCN (Kipf & Welling, 2017), that learns embedding representations of graph nodes. The $l$-th layer of the network is compactly written as

$$H^{(l+1)} = \sigma(\widehat{A}H^{(l)}W^{(l)}), \tag{3}$$

where $\widehat{A}$ is a normalization of the graph adjacency matrix, $W^{(l)}$ is a parameter matrix, and $\sigma$ is a nonlinear activation function. The matrix $H^{(l)}$ contains for each row the embedding of a graph node input to the $l$-th layer, and similarly for the output matrix $H^{(l+1)}$. With $L$ layers, the network transforms an initial feature input matrix $H^{(0)}$ to the output embedding matrix $H^{(L)}$. For a node $v$, the embedding $H^{(L)}(v,:)$ may be fed into a classifier for prediction.

Clearly, in order to compute the gradient of the loss for $v$, one needs the corresponding row of $H^{(L)}$, the rows of $H^{(L-1)}$ corresponding to the neighbors of $v$, and further recursive neighbors across each layer, all the way down to $H^{(0)}$. The computational cost of the unbiased gradient estimator is rather high. In the worst case, all rows of $H^{(0)}$ are involved.

To resolve the inefficiency, Chen et al. (2018) proposed an alternative gradient estimator that is biased but consistent. The simple and effective idea is to sample a constant number of nodes in each layer to restrict the size of the multihop neighborhood. For notational clarity, the approach may be easier to explain for a network with a single layer; theoretical results for more layers straightforwardly follow that of Theorem 1 below, through induction.

The approach generalizes the setting from a finite graph to an infinite graph, such that the matrix expression (3) becomes an integral transform. In particular, the input feature vector $H^{(0)}(u,:)$ for a node $u$ is generalized to a feature function $X(u)$, and the output embedding vector $H^{(1)}(v,:)$ for a node $v$ is generalized to an embedding function $Z(v)$, where the random variables $u$ and $v$ in two sides of the layer reside in different probability spaces, with probability measures $P(u)$ and $P(v)$, respectively. Furthermore, the matrix $\widehat{A}$ is generalized into a bivariate kernel $\widehat{A}(v,u)$ and the loss $\ell$ is written as a function of the output $Z(v)$. Then, (1) and (3) become

$$f = \mathrm{E}_{v \sim P(v)}[\ell(Z(v))] \quad \text{with} \quad Z(v) = \sigma\left(\int \widehat{A}(v,u)X(u)W \, dP(u)\right). \tag{4}$$

Such a functional generalization facilitates sampling on all network layers for defining a gradient estimator. In particular, defining $B(v) = \int \widehat{A}(v,u)X(u) \, dP(u)$, simple calculation reveals that the gradient with respect to the parameter matrix $W$ is

$$G := \nabla f = \int q(B(v)) \, dP(v), \quad \text{where} \quad q(B) = B^T \nabla h(BW) \quad \text{and} \quad h = \ell \circ \sigma.$$

Then, one may use $t$ iid samples of $u$ in the input and $s$ iid samples of $v$ in the output to define an estimator of $G$:

$$G_{st} := \frac{1}{s} \sum_{i=1}^{s} q(B_t(v_i)), \quad v_i \sim P(v), \quad \text{with} \quad B_t(v) := \frac{1}{t} \sum_{j=1}^{t} \widehat{A}(v, u_j)X(u_j), \quad u_j \sim P(u).$$

The gradient estimator $G_{st}$ so defined is consistent; see a proof in the supplementary material.

**Theorem 1.** *If $q$ is continuous and $f$ is finite, then $\mathrm{plim}_{s,t \to \infty} G_{st} = G$.*

## 3 SETTING AND NOTATIONS

We now settle the notations for SGD. We are interested in the (constrained) optimization problem

$$\min_{w \in S} f(w),$$

where the feasible region $S$ is convex. This setting includes the unconstrained case $S = \mathbb{R}^d$. We assume that the objective function $f : \mathbb{R}^d \to \mathbb{R}$ is subdifferentiable; and use $\partial f(w)$ to denote the subdifferential at $w$. When it is necessary to refer to an element of this set, we use the notation $h$. If $f$ is differentiable, then clearly, $\partial f(w) = \{\nabla f(w)\}$.

The standard update rule for SGD is $w_{k+1} = \Pi_S(w_k - \gamma_k g_k)$, where $g_k$ is the negative search direction at step $k$, $\gamma_k$ is the step size, and $\Pi_S$ is the projection onto the feasible region: $\Pi_S(w) := \operatorname{argmin}_{u \in S} \|w - u\|$. For unconstrained problems, the projection is clearly omitted: $w_{k+1} = w_k - \gamma_k g_k$.

Denote by $w^*$ the global minimum. We assume that $w^*$ is an interior point of $S$, so that the subdifferential of $f$ at $w^*$ contains zero. For differentiable $f$, this assumption simply means that $\nabla f(w^*) = 0$.

Typical convergence results are concerned with how fast the iterate $w_k$ approaches $w^*$, or the function value $f(w_k)$ approaches $f(w^*)$. Sometimes, the analysis is made convenient through a convexity assumption on $f$, such that the average of historical function values $f(w_i)$, $i = 1, \ldots, k$, is lowered bounded by $f(\overline{w}_k)$, with $\overline{w}_k$ being the cumulative moving average $\overline{w}_k = \frac{1}{k} \sum_{i=1}^k w_i$.

The following definitions are frequently referenced.

**Definition 3.** We say that $f$ is *l-strongly convex* (with $l > 0$) if for all $w, u \in \mathbb{R}^d$ and $h_u \in \partial f(u)$,

$$f(w) - f(u) \geq \langle h_u, w - u \rangle + \frac{l}{2}\|w - u\|^2.$$

**Definition 4.** We say that $f$ is *L-smooth* (with $L > 0$) if it is differentiable and for all $w, u \in \mathbb{R}^d$,

$$\|\nabla f(w) - \nabla f(u)\| \leq L\|w - u\|.$$

## 4 CONVERGENCE RESULTS

Recall that an estimator $g^N$ of $h$ is consistent if for any $\epsilon > 0$,

$$\lim_{N \to \infty} \Pr(\|g^N - h\| > \epsilon) = 0. \tag{5}$$

In our setting, $h$ corresponds to an element of the subdifferential at step $k$; i.e., $h_k \in \partial f(w_k)$, $g^N$ corresponds to the negative search direction $g_k$, and $N$ corresponds to the sample size $N_k$. That $g_k^{N_k}$ converges to $h_k$ in probability does not imply that $g_k^{N_k}$ is unbiased. Hence, a natural question asks what convergence guarantees exist when using $g_k^{N_k}$ as the gradient estimator. This section answers that question.

First, note that the sample size $N_k$ is associated with not only $g_k^{N_k}$, but also the new iterate $w_{k+1}^{N_k}$. We omit the superscript $N_k$ in these vectors to improve readability.

Similar to the analysis of standard SGD, which is built on the premise of the unbiasedness of $g_k$ and the boundedness of the gradient, in the following subsection we elaborate the parallel assumptions in this work. They are stated only once and will not be repeated in the theorems that follow, to avoid verbosity.

### 4.1 ASSUMPTIONS

The convergence (5) of the estimator does not characterize how fast it approaches the truth. One common assumption is that the probability in (5) decreases exponentially with respect to the sample size. That is, we assume that there exists a step-dependent constant $C_k > 0$ and a nonnegative function $\tau(\delta)$ on the positive axis such that

$$\Pr\left(\|g_k - h_k\| \geq \delta\|h_k\| \,\Big|\, g_1, \ldots, g_{k-1}\right) \leq C_k e^{-N_k \tau(\delta)} \tag{6}$$

for all $k > 1$ and $\delta > 0$. A similar assumption is adopted by Homem-de-Mello (2008) that studied stochastic optimization through sample average approximation. In this case, the exponential tail occurs when the individual moment generating functions exist, a simple application of the Chernoff bound. For the motivating application GCN, the tail is indeed exponential as evidenced by Figure 3.

Note the conditioning on the history $g_1, \ldots, g_{k-1}$ in (6). The reason is that $h_k$ (i.e., the gradient $\nabla f(w_k)$ if $f$ is differentiable) is by itself a random variable dependent on history. In fact, a more rigorous notation for the history should be *filtration*, but we omit the introduction of unnecessary additional definitions here, as using the notion $g_1, \ldots, g_{k-1}$ is sufficiently clear.

**Assumption 1.** *The gradient estimator $g_k$ is consistent and obeys* (6).

The use of a tail bound assumption, such as (6), is to reverse-engineer the required sample size given the desired probability that some event happens. In this particular case, consider the setting where $T$ SGD updates are run. For any $\delta \in (0, 1)$, define the event

$$E_\delta = \Big\{ \|g_1 - h_1\| \leq \delta \|h_1\| \text{ and } \|g_2 - h_2\| \leq \delta \|h_2\| \text{ and } \ldots \text{ and } \|g_T - h_T\| \leq \delta \|h_T\| \Big\}.$$

Given (6) and any $\epsilon \in (0, 1)$, one easily calculates that if the sample sizes satisfy

$$N_k \geq \tau(\delta)^{-1} \log(TC_k/\epsilon), \tag{7}$$

for all $k$, then,

$$\Pr(E_\delta) \geq \prod_{k=1}^{T}(1 - C_k e^{-N_k \tau(\delta)}) \geq \prod_{k=1}^{T}(1 - \epsilon/T) \geq 1 - \epsilon.$$

Hence, all results in this section are established under the event $E_\delta$ that occurs with probability at least $1 - \epsilon$, a sufficient condition of which is (7).

The sole purpose of the tail bound assumption (6) is to establish the relation between the required sample sizes (as a function of $\delta$ and $\epsilon$) and the event $E_\delta$, on which convergence results in this work are based. One may replace the assumption by using other tail bounds as appropriate. It is out of the scope of this work to quantify the rate of convergence of the gradient estimator for a particular use case. For GCN, the exponential tail that agrees with (6) is illustrated in Section 5.4.

Additionally, parallel to the bounded-gradient condition for standard SGD analysis, we impose the following assumption.

**Assumption 2.** *There exists a finite $G > 0$ such that $\|h\| \leq G$ for all $h \in \partial f(w)$ and $w \in S$.*

## 4.2 RESULTS

Let us begin with the strongly convex case. For standard SGD with unbiased gradient estimators, ample results exist that indicate $O(1/T)$ convergence[2] for the expected error, where $T$ is the number of updates; see, e.g., (2.9)–(2.10) of Nemirovski et al. (2009) and Section 3.1 of Lacoste-Julien et al. (2012). We derive similar results for consistent gradient estimators, as stated in the following Theorem 2. Different from the unbiased case, it is the error, rather than the expected error, to be bounded. The tradeoff is the introduction of the relative gradient estimator error $\delta$, which relates to the sample sizes as in (7) for guaranteeing satisfaction of the bound with high probability.

**Theorem 2.** *Let $f$ be $l$-strongly convex with $l \leq G/\|w_1 - w^*\|$. Assume that $T$ updates are run, with diminishing step size $\gamma_k = [(l - \delta)k]^{-1}$ for $k = 1, 2, \ldots, T$, where $\delta = \rho/T$ and $\rho < l$ is an arbitrary constant independent of $T$. Then, for any such $\rho$, any $\epsilon \in (0, 1)$, and sufficiently large sample sizes satisfying (7), with probability at least $1 - \epsilon$, we have*

$$\|w_T - w^*\|^2 \leq \frac{G^2}{T}\left[\frac{(1 + \rho/T)^2 + \rho(l - \rho/T)}{(l - \rho/T)^2}\right], \tag{8}$$

*and*

$$f(\overline{w}_T) - f(w^*) \leq \frac{G^2}{2T}\left[\rho + \frac{(1 + \rho/T)^2}{l - \rho/T}(1 + \log T)\right]. \tag{9}$$

---

[2]Ignoring the logarithmic factor, if any.

Note the assumption on $l$ in Theorem 2. This assumption is mild since if $f$ is $l$-strongly convex, it is also $l'$-strongly convex for all $l' < l$. The assumption is needed in the induction proof of (8) when establishing the base case $\|w_1 - w^*\|$. One may remove this assumption at the cost of a cumbersome right-hand side of (8), over which we favor a neater expression in the current form.

With an additional smoothness assumption, we may eliminate the logarithmic factor in (9) and obtain a result for the iterate $w_T$ rather than the running average $\overline{w}_T$. The result is a straightforward consequence of (8).

**Theorem 3.** *Under the conditions of Theorem 2, additionally let $f$ be $L$-smooth. Then, for any $\rho$ satisfying the conditions, any $\epsilon \in (0,1)$, and sufficiently large sample sizes satisfying (7), with probability at least $1 - \epsilon$, we have*

$$f(w_T) - f(w^*) \leq \frac{LG^2}{2T} \left[ \frac{(1 + \rho/T)^2 + \rho(l - \rho/T)}{(l - \rho/T)^2} \right]. \tag{10}$$

In addition to $O(1/T)$ convergence, it is also possible to establish linear convergence (however) to a non-vanishing right-hand side, as the following result indicates. To obtain such a result, we use a constant step size. Bottou et al. (2016) show a similar result for the function value with an additional smoothness assumption in a different setting; we give one for the iterate error without the smoothness assumption using consistent gradients.

**Theorem 4.** *Under the conditions of Theorem 2, except that one sets a constant step size $\gamma_k = c$ with $0 < c < (2l - \delta)^{-1}$ for all $k$, for any $\rho$ satisfying the conditions, any $\epsilon \in (0,1)$, and sufficiently large sample sizes satisfying (7), with probability at least $1 - \epsilon$, we have*

$$\|w_T - w^*\|^2 \leq (1 - 2cl + c\delta)^{T-1} \|w_1 - w^*\|^2 + \frac{\delta + c(1 + \delta)^2}{2l - \delta} G^2. \tag{11}$$

Compare (11) with (8) in Theorem 2. The former indicates that in the limit, the squared iterate error is upper bounded by a positive term proportional to $G^2$; the remaining part of this upper bound decreases at a linear speed. The latter, on the other hand, indicates that the squared iterate error in fact will vanish, although it does so at a sublinear speed $O(1/T)$.

For convex (but not strongly convex) $f$, typically $O(1/\sqrt{T})$ convergence is asserted for unbiased gradient estimators; see., e.g., Theorem 2 of Liu (2015). These results are often derived based on an additional assumption that the feasible region is compact. Such an assumption is not restrictive, because even if the problem is unconstrained, one can always confine the search to a bounded region (e.g., an Euclidean ball). Under this condition, we obtain a similar result for consistent gradient estimators.

**Theorem 5.** *Let $f$ be convex and the feasible region $S$ have finite diameter $D > 0$; that is, $\sup_{w,u \in S} \|w - u\| = D$. Assume that $T$ updates are run, with diminishing step size $\gamma_k = c/\sqrt{k}$ for $k = 1, 2, \ldots, T$ and for some $c > 0$. Let $\delta = \rho/\sqrt{T}$ where $\rho > 0$ is an arbitrary constant independent of $T$. Then, for any such $\rho$, any $\epsilon \in (0,1)$, and sufficiently large sample sizes satisfying (7), with probability at least $1 - \epsilon$, we have*

$$f(\overline{w}_T) - f(w^*) \leq \frac{1}{2\sqrt{T}} \left[ \left( \frac{1}{c} + \rho \right) D^2 + G^2 \left( \rho + c \left( 1 + \frac{\rho}{\sqrt{T}} \right)^2 \sqrt{1 + \frac{1}{T}} \right) \right]. \tag{12}$$

One may obtain a result of the same convergence rate by using a constant step size. In the case of unbiased gradient estimators, see Theorem 14.8 of Shalev-Shwartz & Ben-David (2014). For such a result, one assumes that the step size is inversely proportional to $\sqrt{T}$. Such choice of the step size is common and is also used in the next setting.

For the general (nonconvex) case, convergence is typically gauged with the gradient norm. One again obtains $O(1/\sqrt{T})$ convergence results for unbiased gradient estimators; see, e.g., Theorem 1 of Reddi et al. (2016) (which is a simplified consequence of the theory presented in Ghadimi & Lan (2013)). We derive a similar result for consistent gradient estimators.

**Theorem 6.** *Let $f$ be $L$-smooth and $S = \mathbb{R}^d$. Assume that $T$ updates are run, with constant step size $\gamma_k = D_f/[(1 + \delta)G\sqrt{T}]$ for $k = 1, 2, \ldots, T$, where $D_f = [2(f(w_1) - f(w^*))/L]^{\frac{1}{2}}$, and $\delta \in (0,1)$*

*is an arbitrary constant. Then, for any such $\delta$, any $\epsilon \in (0, 1)$, and sufficiently large sample sizes satisfying* (7)*, with probability at least $1 - \epsilon$, we have*

$$\min_{k=1,\dots,T} \|\nabla f(w_k)\|^2 \leq \frac{(1+\delta)LGD_f}{(1-\delta)\sqrt{T}}. \tag{13}$$

### 4.3 INTERPRETATION

All the results in the preceding subsection assert convergence for SGD with the use of a consistent gradient estimator. As with the use of an unbiased one, the convergence for the strongly convex case is $O(1/T)$, or linear if one tolerates a non-vanishing upper bound, and the convex and nonconvex cases $O(1/\sqrt{T})$. These theoretical results, however, are based on assumptions of the sample size $N_k$ and the step size $\gamma_k$ that are practically challenging to verify. Hence, in a real-life machine learning setting, the sample size and the learning rate (the initial step size) are treated as hyperparameters to be tuned against a validation set.

Nevertheless, these results establish a qualitative relationship between the sample size and the optimization error. Naturally, to maintain the same failure probability $\epsilon$, the relative gradient estimator error $\delta$ decreases inversely with the sample size $N_k$. This intuition holds true in the tail bound condition (6) with (7), when $\tau(\delta)$ is a monomial or a positive combination of monomials with different degrees. With this assumption, the larger is $N_k$, the smaller is $\delta$ (and also $\rho$, the auxiliary quantity defined in the theorems); hence, the smaller are the error bounds (8)–(13).

### 4.4 REMARKS

Theorem 4 presents a linear convergence result for the strongly convex case, with a non-vanishing right-hand side. In fact, it is possible to obtain a result with the same convergence rate but a vanishing right-hand side, if one is willing to additionally assume $L$-smoothness. The following theorem departs from the set of theorems in Section 4.2 on the assumption of the sufficient sample size $N_k$ and the gradient error $\delta$.

**Theorem 7.** *Let $f$ be $l$-strongly convex and $L$-smooth with $l < L$. Assume that $T$ updates are run with constant step size $\gamma_k = 1/L$ for $k = 1, 2, \dots, T$. Let $\delta_k$, $k \geq 1$ be a sequence where $\lim_{k\to\infty} \delta_{k+1}/\delta_k \leq 1$. Then, for any positive $\eta < l/L$, $\epsilon \in (0, 1)$, and sample sizes*

$$N_k \geq \tau(\delta_k/\|h_k\|)^{-1} \log(TC_k/\epsilon) \quad \text{for } k = 1, 2, \dots, T,$$

*with probability at least $1 - \epsilon$, we have*

$$f(w_T) - f(w^*) \leq (1 - l/L)^{T-1}[f(w_1) - f(w^*)] + O(E_T), \tag{14}$$

*where $E_T = \max\{\delta_T^2, (1 - l/L + \eta)^T\}$.*

Here, $\delta_k$ is the step-dependent gradient error. If it decreases to zero, then so does $E_T$. Theorem 7 is adapted from Friedlander & Schmidt (2012), who studied unbiased gradients as well as noisy gradients. We separate Theorem 7 from those in Section 4.2 only for the sake of presentation clarity. The spirit, however, remains the same. Namely, consistent estimators result in the same convergence behavior (i.e., rate) as do unbiased ones. All results require an assumption on sufficient sample size owing to the probabilistic convergence of the gradient estimator.

## 5 NUMERICAL ILLUSTRATIONS

In this section, we report several experiments to illustrate the convergence behavior of SGD by using consistent gradient estimators. We base the experiments on the training of the GCN model (Kipf & Welling, 2017) motivated earlier (cf. Section 2). The code repository will be revealed upon paper acceptance.

### 5.1 DATA SETS

We use three data sets for illustration, one synthetic and two real-world benchmarks.

The purpose of a synthetic data set is to avoid the regularity in the sampling of training/validation/test examples. The data set, called "Mixture," is a mixture of three overlapping Gaussians. The points are randomly connected, with a higher probability for those within the same component than the ones straddling across components. See the supplementary material for details of the construction. Because of the significant overlap, a classifier trained with independent data points unlikely predicts well the component label, but a graph-based method is more likely to be successful.

Additionally, we use two benchmark data sets, Cora and Pubmed, often seen in the literature. These graphs are citation networks and the task is to predict the topics of the publications. We follow the split used in Chen et al. (2018). See the supplementary material for a summary of all data sets.

## 5.2 (STRONGLY) CONVEX CASE

The GCN model is hyperparameterized by the number of layers. Without any intermediate layer, the model can be considered a generalized linear model and thus the cross-entropy loss function is convex. Moreover, with the use of an $L_2$ regularization, the loss becomes strongly convex. The predictive model reads $P = \text{softmax}(\widehat{A}XW^{(0)})$, where $X$ is the input feature matrix and $P$ is the output probability matrix, both row-wise. One easily sees that the only difference between this model and logistic regression $P = \text{softmax}(XW^{(0)})$ is the neighborhood aggregation $\widehat{A}X$.

Standard batched training in SGD samples a batch (denoted by the index set $I_1$) from the training set and evaluates the gradient of the loss of $\text{softmax}(\widehat{A}(I_1, :)XW^{(0)})$. In the analyzed consistent-gradient training, we additionally uniformly sample the input layer with another index set $I_0$ and evaluate instead the gradient of the loss of $\text{softmax}(\frac{n}{|I_0|}\widehat{A}(I_1, I_0)X(I_0, :)W^{(0)})$.

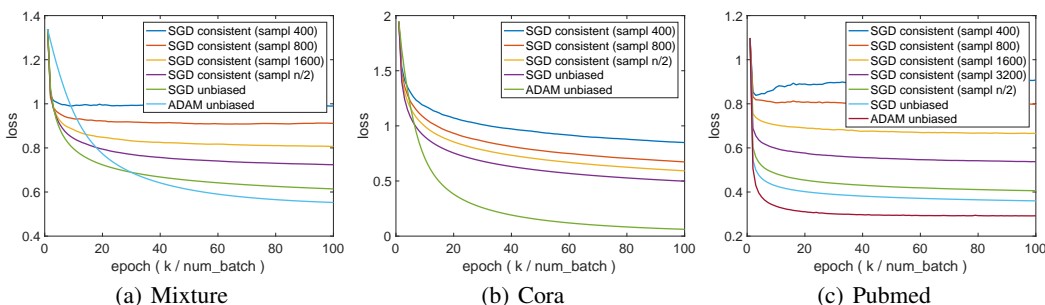

Figure 1: Convergence history for 1-layer GCN, under different training algorithms.

Figure 1 shows the convergence curves as the iteration progresses. The plotted quantity is the overall loss on all training examples, rather than the batch loss for only the current batch. Hence, not surprisingly the curves are generally quite smooth. We compare standard SGD with the use of consistent gradient estimators, with varying sample size $|I_0|$. Additionally, we compare with the Adam training algorithm (Kingma & Ba, 2015), which is a stochastic optimization approach predominantly used in practice for training deep neural networks.

One sees that for all data sets, Adam converges faster than does standard SGD. Moreover, as the sample size increases, the loss curve with consistent gradients approaches that with an unbiased one (i.e., standard SGD). This phenomenon qualitatively agrees with the theoretical results; namely, larger sample size improves the error bound. Note that all curves in the same plot result from the same parameter initialization; and all SGD variants apply the same learning rate.

It is important to note that the training loss is only a surrogate measure of the model performance; and often early termination of the optimization acts as a healthy regularization against over-fitting. In our setting, a small sample size may not satisfy the assumptions of the theoretical results, but it proves to be practically useful. In Table 1 (left), we report the test accuracy attained by different training algorithms at the epoch where validation accuracy peaks. One sees that Adam and standard SGD achieves similar accuracies, and that SGD with consistent gradient sometimes surpasses these accuracies. For Cora, a sample size 400 already yields an accuracy noticeably higher than do Adam and standard SGD.

Table 1: Test accuracy (in percentage) and epoch number (inside parentheses) for different GCN architectures and training algorithms. For the same architecture, initialization is the same. The epoch number is the one when best validation accuracy occurs.

| | 1-layer GCN | | | 2-layer GCN | | |
|---|---|---|---|---|---|---|
| | Mixture | Cora | Pubmed | Mixture | Cora | Pubmed |
| SGD (400) | 78.0 (68) | 85.8 (97) | 86.2 (15) | 86.7 (76) | 87.1 (34) | 87.5 (88) |
| SGD (800) | 77.8 (46) | 86.1 (86) | 87.9 (68) | 86.9 (87) | 85.8 (13) | 87.6 (87) |
| SGD (1600) | 77.9 (87) | - | 88.6 (35) | 86.8 (94) | - | 88.3 (85) |
| SGD (3200) | - | - | 88.9 (98) | - | - | 88.1 (88) |
| SGD unbiased | 78.1 (93) | 84.2 (87) | 88.1 (75) | 86.8 (66) | 87.4 (27) | 87.9 (90) |
| Adam unbiased | 80.0 (95) | 84.9 (21) | 88.4 (20) | 87.6 (94) | 87.0 (04) | 88.0 (06) |

## 5.3 NONCONVEX CASE

When GCN has intermediate layers, the loss function is generally nonconvex. A 2-layer GCN reads $P = \text{softmax}(\widehat{A} \cdot \text{ReLU}(\widehat{A}XW^{(0)}) \cdot W^{(1)})$, and a GCN with more layers is analogous.

We repeat the experiments in the preceding subsection. The results are reported in Figure 2 and Table 1 (right). The observation of the loss curve follows the same as that in the convex case. Namely, Adam converges faster than does unbiased SGD; and the convergence curve with a consistent gradient approaches that with an unbiased one.

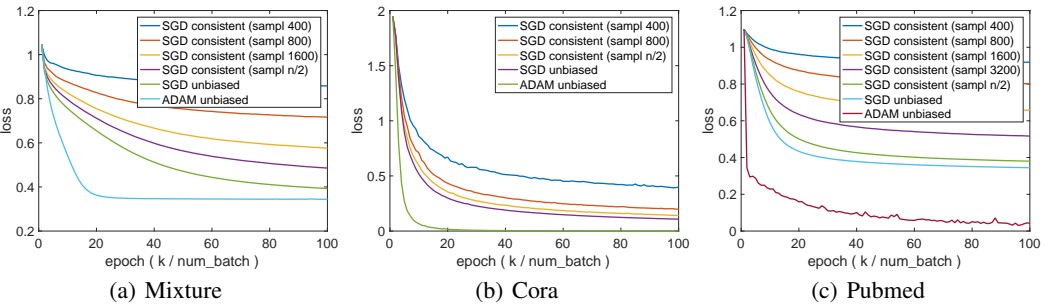

(a) Mixture        (b) Cora        (c) Pubmed

Figure 2: Convergence history for 2-layer GCN, under different training algorithms.

On the other hand, compared with 1-layer GCN, 2-layer GCN yields substantially higher test accuracy for the data set Mixture, better accuracy for Cora, and very similar accuracy for Pubmed. Within each data set, the performances of different training algorithms are on par. In particular, a small sample size (e.g., 400) suffices for achieving results comparable to the state of the art (cf. Chen et al. (2018)).

## 5.4 PROBABILITY CONVERGENCE

The nature of a consistent estimator necessitates a characterization of the speed of probability convergence for building further results, such as the ones in this paper. The speed, however, depends on the neural network architecture and it is out of the scope of this work to quantify it for a particular use case. Nevertheless, for GCN we demonstrate empirical findings that agree with the exponential tail assumption (6). In Figure 3 (solid curves), we plot the tail probability as a function of the sample size $N$ at different levels of estimator error $\delta$, for the initial gradient step in 1-layer GCN. For each $N$, 10,000 random gradient estimates were simulated for estimating the probability. Because the probability is plotted in the logarithmic scale, the fact that the curves bend down indicates that the convergence may be faster than exponential.

Additionally, the case of 2-layer GCN is demonstrated by the dashed curves in Figure 3. The curves tend to be straight lines in the limit, which indicates an exponential convergence.

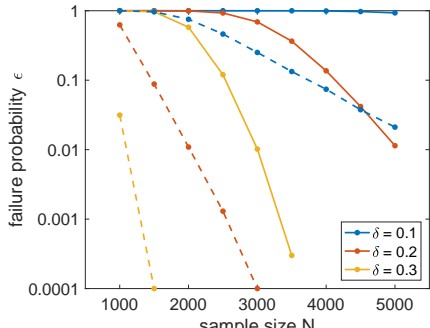

Figure 3: Failure probability versus sample size at different levels of estimator error $\delta$. Solid: 1-layer GCN; dashed: 2-layer GCN.

## 6 CONCLUDING REMARKS

To the best of our knowledge, this is the first work that studies the convergence behavior of SGD with consistent gradient estimators, and one among few studies of first-order methods that employ biased (d'Aspremont, 2008; Schmidt et al., 2011) or noisy (Friedlander & Schmidt, 2012; Devolder et al., 2014; Ge et al., 2015) estimators. The motivation originates from learning with large graphs and the main message is that the convergence behavior is well-maintained with respect to the unbiased case. While we analyze the classic SGD update formula, this work points to several immediate extensions. One direction is the design of more efficient update formulas resembling the variance reduction technique for unbiased estimators (Johnson & Zhang, 2013; Defazio et al., 2014; Bottou et al., 2016). Another direction is the development of more computation- and memory-efficient training algorithms for neural networks for large graphs. GCN is only one member of a broad family of message passing neural networks (Gilmer et al., 2017) that suffer from the same limitation of neighborhood aggregation. Learning in these cases inevitably faces the costly computation of the sample gradient. Hence, a consistent estimator appears to be a promising alternative, whose construction is awaiting more innovative proposals.

We are grateful to an anonymous reviewer who inspired us of an interesting use case (other than GCN). *Learning to rank* is a machine learning application that constructs ranking models for information retrieval systems. In representative methods such as RankNet (Burges et al., 2005) and subsequent improvements (Burges et al., 2007; Burges, 2010), $s_i$ is the ranking function for document $i$ and the learning amounts to minimizing the loss

$$\sum_{(i,j)} s_j - s_i + \log(1 + e^{s_i - s_j}),$$

where the summation ranges over all pairs of documents such that $i$ is ranked higher than $j$. The pairwise information may be organized as a graph and the loss function may be similarly generalized as a double integral analogous to (4). Because of nonlinearity, Monte Carlo sampling of each integral will result in a biased but consistent estimator. Therefore, a new training algorithm is to sample $i$ and $j$ separately (forming a consistent gradient) and apply SGD. The theory developed in this work offers guarantees of training convergence.

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

# A  PROOFS

## A.1  LEMMAS

Here are a few lemmas needed for the proofs in subsequent subsections.

**Lemma 8.** *Projection is nonexpanding, i.e.,*

$$\|\Pi_S(w) - \Pi_S(u)\| \le \|w - u\|, \quad \forall\, w, u \in \mathbb{R}^d.$$

*Proof.* Let $w' = \Pi_S(w)$ and $u' = \Pi_S(u)$. By the convexity of $S$, we have

$$\langle w - w', u' - w' \rangle \le 0 \quad \text{and} \quad \langle u - u', w' - u' \rangle \le 0.$$

Summing these two inequalities, we obtain $\langle w - u, w' - u' \rangle \ge \langle w' - u', w' - u' \rangle$. Then, by Cauchy–Schwarz,

$$\|w' - u'\|^2 \le \langle w - u, w' - u' \rangle \le \|w - u\|\|w' - u'\|,$$

which concludes the proof.  □

**Lemma 9.** *If $f$ is $l$-strongly convex, then*

$$\langle h_u, u - w^* \rangle \ge l\|u - w^*\|^2, \quad \forall\, u \in \mathbb{R}^d \text{ and } h_u \in \partial f(u).$$

*Proof.* Applying Definition 3 twice

$$f(w^*) - f(u) \ge \langle h_u, w^* - u \rangle + \frac{l}{2}\|w^* - u\|^2$$

$$f(u) - f(w^*) \ge \qquad\qquad\quad + \frac{l}{2}\|u - w^*\|^2,$$

and summing these two inequalities, we conclude the proof.  □

**Lemma 10.** *For any $w \in S$,*

$$\|w_{k+1} - w\|^2 \le \|w_k - w\|^2 - 2\gamma_k \langle g_k, w_k - w \rangle + \gamma_k^2 \|g_k\|^2.$$

*Proof.* It is straightforward to verify that

$$\begin{aligned}
\|w_{k+1} - w\|^2 &= \|\Pi_S(w_k - \gamma_k g_k) - w\|^2 \\
&\le \|w_k - \gamma_k g_k - w\|^2 \\
&= \|w_k - w\|^2 - 2\gamma_k \langle g_k, w_k - w \rangle + \gamma_k^2 \|g_k\|^2,
\end{aligned}$$

where the inequality results from Lemma 8.  □

**Lemma 11.** *If $\|g_k - h_k\| \le \delta\|h_k\|$, then*

$$(1 - \delta)\|h_k\| \le \|g_k\| \le (1 + \delta)\|h_k\|,$$

*and*

$$-\frac{\delta}{2}(\|h_k\|^2 + \|w_k - w^*\|^2) \le \langle g_k - h_k, w_k - w^* \rangle \le \frac{\delta}{2}(\|h_k\|^2 + \|w_k - w^*\|^2).$$

*Proof.* For the first displayed inequality, it is straightforward to verify the upper bound

$$\|g_k\| \leq \|h_k\| + \|g_k - h_k\| \leq (1 + \delta)\|h_k\|,$$

and similarly the lower bound. For the second displayed inequality, Cauchy–Schwarz leads to the upper bound

$$\langle g_k - h_k, w_k - w^* \rangle \leq \|g_k - h_k\| \cdot \|w_k - w^*\| \leq \delta\|h_k\| \cdot \|w_k - w^*\| \leq \frac{\delta}{2}(\|h_k\|^2 + \|w_k - w^*\|^2).$$

The lower bound is similarly proved. □

## A.2 PROOF OF THEOREM 1

By the weak law of large numbers, $B_t(v) \to B(v)$ in probability for any $v$, where the probability space is with respect to $u$. Then, $q(B_t(v)) \to q(B(v))$ in probability by the continuous mapping theorem. Applying the law of large numbers again, now for $v$ on a separate probability space different from that of $u$, we conclude that $G_{st} \to G$ in probability.

## A.3 PROOF OF THEOREM 2, INEQUALITY (8)

Applying Lemma 10 with $w = w^*$, we have

$$\|w_{k+1} - w^*\|^2 \leq \|w_k - w^*\|^2 - 2\gamma_k \langle h_k, w_k - w^* \rangle - 2\gamma_k \langle g_k - h_k, w_k - w^* \rangle + \gamma_k^2 \|g_k\|^2.$$

Applying Lemma 9 with $u = w_k$ and Lemma 11, we have

$$
\begin{aligned}
\|w_{k+1} - w^*\|^2 &\leq \|w_k - w^*\|^2 - 2\gamma_k l\|w_k - w^*\|^2 + \gamma_k \delta(\|h_k\|^2 + \|w_k - w^*\|^2) + \gamma_k^2(1 + \delta)^2\|h_k\|^2 \\
&= (1 - 2\gamma_k l + \gamma_k \delta)\|w_k - w^*\|^2 + (\gamma_k \delta + \gamma_k^2(1 + \delta)^2)G^2.
\end{aligned} \tag{15}
$$

In what follows, we show by induction on $k$ that

$$\|w_k - w^*\|^2 \leq \left[\frac{(1 + \delta)^2}{(l - \delta)^2 k} + \frac{\delta}{l - \delta}\right]G^2.$$

Then, setting $k = T$ we can conclude the proof.

First, in the base case when $k = 1$, by assumption we have

$$\|w_k - w^*\|^2 \leq \frac{G^2}{l^2} \leq \frac{(1 + \delta)^2}{(l - \delta)^2}G^2 \leq \left[\frac{(1 + \delta)^2}{(l - \delta)^2} + \frac{\delta}{l - \delta}\right]G^2.$$

Then, in the induction step, taking $\gamma_k = [(l - \delta)k]^{-1}$ as defined in the theorem on (15) and using the induction hypothesis, we have

$$
\begin{aligned}
\|w_{k+1} - w^*\|^2 &\leq \frac{lk - \delta k - 2l + \delta}{(l - \delta)k}\left[\frac{(1 + \delta)^2}{(l - \delta)^2 k} + \frac{\delta}{l - \delta}\right]G^2 + \left[\frac{\delta}{(l - \delta)k} + \frac{(1 + \delta)^2}{(l - \delta)^2 k^2}\right]G^2 \\
&= \frac{(lk - \delta k - 2l + \delta)(1 + \delta)^2}{(l - \delta)^3 k^2}G^2 + \frac{(lk - \delta k - 2l + \delta)\delta}{(l - \delta)^2 k}G^2 + \frac{\delta}{(l - \delta)k}G^2 + \frac{(1 + \delta)^2}{(l - \delta)^2 k^2}G^2 \\
&\leq \frac{(k - 2)(1 + \delta)^2}{(l - \delta)^2 k^2}G^2 + \frac{\delta(k - 1)}{(l - \delta)k}G^2 + \frac{\delta}{(l - \delta)k}G^2 + \frac{(1 + \delta)^2}{(l - \delta)^2 k^2}G^2.
\end{aligned}
$$

For the right-hand side, combine the first and the fourth term, and the second and the third term, we obtain

$$\|w_{k+1} - w^*\|^2 \leq \frac{(k - 1)(1 + \delta)^2}{(l - \delta)^2 k^2}G^2 + \frac{\delta}{(l - \delta)}G^2 \leq \frac{(1 + \delta)^2}{(l - \delta)^2(k + 1)}G^2 + \frac{\delta}{(l - \delta)}G^2,$$

which completes the induction step.

### A.4  PROOF OF THEOREM 2, INEQUALITY (9)

Applying Lemma 10 with $w = w^*$, we have

$$\|w_{k+1} - w^*\|^2 \leq \|w_k - w^*\|^2 - 2\gamma_k \langle h_k, w_k - w^* \rangle - 2\gamma_k \langle g_k - h_k, w_k - w^* \rangle + \gamma_k^2 \|g_k\|^2.$$

Applying the definition of strong convexity and Lemma 11, we have

$$\|w_{k+1} - w^*\|^2 \leq \|w_k - w^*\|^2 - 2\gamma_k[f(w_k) - f(w^*)] - \gamma_k l\|w_k - w^*\|^2$$
$$+ \gamma_k \delta(\|h_k\|^2 + \|w_k - w^*\|^2) + \gamma_k^2(1+\delta)^2\|h_k\|^2$$
$$= -2\gamma_k[f(w_k) - f(w^*)] + (1 - \gamma_k l + \gamma_k \delta)\|w_k - w^*\|^2 + (\gamma_k \delta + \gamma_k^2(1+\delta)^2)G^2.$$

Rearranging, we have

$$2[f(w_k) - f(w^*)] \leq (\gamma_k^{-1} - l + \delta)\|w_k - w^*\|^2 - \gamma_k^{-1}\|w_{k+1} - w^*\|^2 + (\delta + \gamma_k(1+\delta)^2)G^2.$$

Noting that the step size $\gamma_k = [(l-\delta)k]^{-1}$, we have

$$2[f(w_k) - f(w^*)] \leq (l-\delta)(k-1)\|w_k - w^*\|^2 - (l-\delta)k\|w_{k+1} - w^*\|^2 + G^2\left[\delta + \frac{(1+\delta)^2}{(l-\delta)k}\right].$$

Summing from $k = 1$ to $k = T$ and multiplying by $1/(2T)$, we have

$$\frac{1}{T}\sum_{k=1}^{T}[f(w_k) - f(w^*)] \leq -\frac{l-\delta}{2}\|w_{T+1} - w^*\|^2 + \frac{G^2}{2T}\left[\delta T + \frac{(1+\delta)^2}{l-\delta}\sum_{k=1}^{T}\frac{1}{k}\right].$$

By the convexity of $f$ and using the bound $\sum_{k=1}^{T} 1/k \leq 1 + \log T$, and noting that $\delta = \rho/T$, we have

$$f(\overline{w}_T) - f(w^*) \leq -\frac{l-\delta}{2}\|w_{T+1} - w^*\|^2 + \frac{G^2}{2T}\left[\rho + \frac{(1+\rho/T)^2}{l-\rho/T}(1+\log T)\right].$$

Relaxing the right-hand side through omitting the negative term, we thus conclude the proof.

### A.5  PROOF OF THEOREM 3

The $L$-smoothness property implies a second order condition for convex functions:

$$f(w_k) - f(w^*) \leq \frac{L}{2}\|w_k - w^*\|^2.$$

Then, applying (8) with $k = T$, we conclude the proof.

### A.6  PROOF OF THEOREM 4

We reuse (15) in the proof of inequality (8) in Theorem 2:

$$\|w_{k+1} - w^*\|^2 \leq (1 - 2\gamma_k l + \gamma_k \delta)\|w_k - w^*\|^2 + (\gamma_k \delta + \gamma_k^2(1+\delta)^2)G^2.$$

Applying step size $\gamma_k = c$, we have

$$\|w_{k+1} - w^*\|^2 \leq (1 - 2cl + c\delta)\|w_k - w^*\|^2 + (c\delta + c^2(1+\delta)^2)G^2.$$

Unrolling recursion with respect to $k$, we have

$$\|w_{k+1} - w^*\|^2 \leq (1 - 2cl + c\delta)^k\|w_1 - w^*\|^2 + (c\delta + c^2(1+\delta)^2)\sum_{i=0}^{k-1}(1 - 2cl + c\delta)^i G^2.$$

Because $0 < 1 - 2cl + c\delta < 1$ by assumption, we have

$$\sum_{i=0}^{k-1}(1 - 2cl + c\delta)^i < \frac{1}{2cl - c\delta},$$

which concludes the proof.

### A.7 PROOF OF THEOREM 5

Applying Lemma 10 with $w = w^*$, we have

$$\|w_{k+1} - w^*\|^2 \le \|w_k - w^*\|^2 - 2\gamma_k \langle h_k, w_k - w^* \rangle - 2\gamma_k \langle g_k - h_k, w_k - w^* \rangle + \gamma_k^2 \|g_k\|^2.$$

Applying a property of convex functions and Lemma 11, we have

$$\|w_{k+1} - w^*\|^2 \le \|w_k - w^*\|^2 - 2\gamma_k[f(w_k) - f(w^*)] + \gamma_k\delta(\|h_k\|^2 + \|w_k - w^*\|^2) + \gamma_k^2(1+\delta)^2\|h_k\|^2$$
$$= -2\gamma_k[f(w_k) - f(w^*)] + (1 + \gamma_k\delta)\|w_k - w^*\|^2 + (\gamma_k\delta + \gamma_k^2(1+\delta)^2)G^2.$$

Rearranging, we have

$$2[f(w_k) - f(w^*)] \le (\gamma_k^{-1} + \delta)\|w_k - w^*\|^2 - \gamma_k^{-1}\|w_{k+1} - w^*\|^2 + (\delta + \gamma_k(1+\delta)^2)G^2.$$

Summing from $k = 1$ to $k = T$, relaxing the negative term $-\gamma_T^{-1}\|w_{T+1} - w^*\|^2$ on the right-hand side, and multiplying by $1/(2T)$, we have

$$\frac{1}{T}\sum_{k=1}^{T}[f(w_k) - f(w^*)] \le \frac{\gamma_1^{-1} + \delta}{2T}\|w_1 - w^*\|^2$$
$$+ \sum_{k=2}^{T}\frac{\gamma_k^{-1} + \delta - \gamma_{k-1}^{-1}}{2T}\|w_k - w^*\|^2 + \frac{G^2}{2T}\left[\delta T + (1+\delta)^2\sum_{k=1}^{T}\gamma_k\right].$$

Applying $\|w_k - w^*\|^2 \le D^2$ for all $k$, we have

$$\frac{1}{T}\sum_{k=1}^{T}[f(w_k) - f(w^*)] \le \frac{(\gamma_T^{-1} + \delta T)D^2}{2T} + \frac{G^2}{2T}\left[\delta T + (1+\delta)^2\sum_{k=1}^{T}\gamma_k\right].$$

Noting that $\gamma_k = c/\sqrt{k}$ and $\delta = \rho/\sqrt{T}$, we have

$$\frac{1}{T}\sum_{k=1}^{T}[f(w_k) - f(w^*)] \le \frac{1}{2\sqrt{T}}\left(\frac{1}{c} + \rho\right)D^2 + \frac{G^2}{2T}\left[\rho\sqrt{T} + c\left(1 + \frac{\rho}{\sqrt{T}}\right)^2\sum_{k=1}^{T}\frac{1}{\sqrt{k}}\right].$$

By the convexity of $f$ and using the bound $\sum_{k=1}^{T} 1/\sqrt{k} \le \sqrt{T+1}$, we have

$$f(\overline{w}_T) - f(w^*) \le \frac{1}{2\sqrt{T}}\left(\frac{1}{c} + \rho\right)D^2 + \frac{G^2}{2T}\left[\rho\sqrt{T} + c\left(1 + \frac{\rho}{\sqrt{T}}\right)^2\sqrt{T+1}\right],$$

which concludes the proof.

### A.8 PROOF OF THEOREM 6

The $L$-smoothness property implies that

$$f(w_{k+1}) \le f(w_k) + \langle \nabla f(w_k), w_{k+1} - w_k \rangle + \frac{L}{2}\|w_{k+1} - w_k\|^2.$$

Noting that $w_{k+1} - w_k = -\gamma_k g_k$ (because $S = \mathbb{R}^d$) and applying Lemma 11, we have

$$f(w_{k+1}) \le f(w_k) - \gamma_k\langle h_k, g_k \rangle + \frac{L\gamma_k^2\|g_k\|^2}{2} \le f(w_k) - \gamma_k(1-\delta)\|h_k\|^2 + \frac{L\gamma_k^2(1+\delta)^2\|h_k\|^2}{2}.$$

Rearranging, we have

$$\|\nabla f(w_k)\|^2 \le [\gamma_k(1-\delta)]^{-1}[f(w_k) - f(w_{k+1})] + \frac{L\gamma_k(1+\delta)^2G^2}{2(1-\delta)}.$$

Summing from $k = 1$ to $k = T$, multiplying by $1/T$, and noting that $\gamma_k$ is constant, we have

$$\min_k \|\nabla f(w_k)\|^2 \le \frac{[\gamma_1(1-\delta)]^{-1}}{T}[f(w_1) - f(w_{T+1})] + \frac{L\gamma_1(1+\delta)^2G^2}{2(1-\delta)}.$$

Because $f(w_{T+1}) \ge f(w^*)$ and $\gamma_1 = D_f/[(1+\delta)G\sqrt{T}]$, we have

$$\min_k \|\nabla f(w_k)\|^2 \le \frac{[\gamma_1(1-\delta)]^{-1}}{T}[f(w_1) - f(w^*)] + \frac{L\gamma_1(1+\delta)^2G^2}{2(1-\delta)} = \frac{(1+\delta)LGD_f}{(1-\delta)\sqrt{T}},$$

which concludes the proof.

## A.9 PROOF OF THEOREM 7

Theorem 2.2 of Friedlander & Schmidt (2012) states that when the gradient error

$$\|g_k - h_k\| < \delta_k \quad \text{for all} \quad k \geq 1, \tag{16}$$

inequality (14) holds. It remains to show that the probability that (16) happens is at least $1 - \epsilon$.

The assumption on the sample size $N_k$ means that

$$C_k e^{-N_k \tau(\delta_k/\|h_k\|)} \leq \epsilon/T.$$

Then, substituting $\delta_k = \delta\|h_k\|$ into assumption (6) yields

$$\Pr\left(\|g_k - h_k\| \geq \delta_k \,\Big|\, g_1, \ldots, g_{k-1}\right) \leq C_k e^{-N_k \tau(\delta_k/\|h_k\|)} \leq \epsilon/T.$$

Hence, the probability that (16) happens is

$$\prod_{k=1}^{T}\left[1 - \Pr\left(\|g_k - h_k\| \geq \delta_k \,\Big|\, g_1, \ldots, g_{k-1}\right)\right] \geq \prod_{k=1}^{T}(1 - \epsilon/T) \geq 1 - \epsilon,$$

which concludes the proof.

## B  EXPERIMENT DETAILS

### B.1  THE "MIXTURE" DATA SET

The data set is a Gaussian mixture with $c = 3$ components in $d = 2$ dimensions. The components $\mathcal{N}(\mu_i, \sigma_i^2 I)$ with $\mu_1 = [-0.5, 0]$, $\sigma_1 = 0.75$, $\mu_2 = [0.5, 0]$, $\sigma_2 = 0.5$, $\mu_3 = [0, 0.866]$, and $\sigma_3 = 0.25$ are equally weighted but significantly overlap with each other. Random connections are made between every pair of points. For points in the same component, the probability that they are connected is $p_{\text{intra}} = 1\text{e-}3$; for points straddle across components, the probability is $p_{\text{inter}} = 2\text{e-}4$. See Figure 4(a) for an illustration of the Gaussian mixture and Figure 4(b) for the graph adjacency matrix.

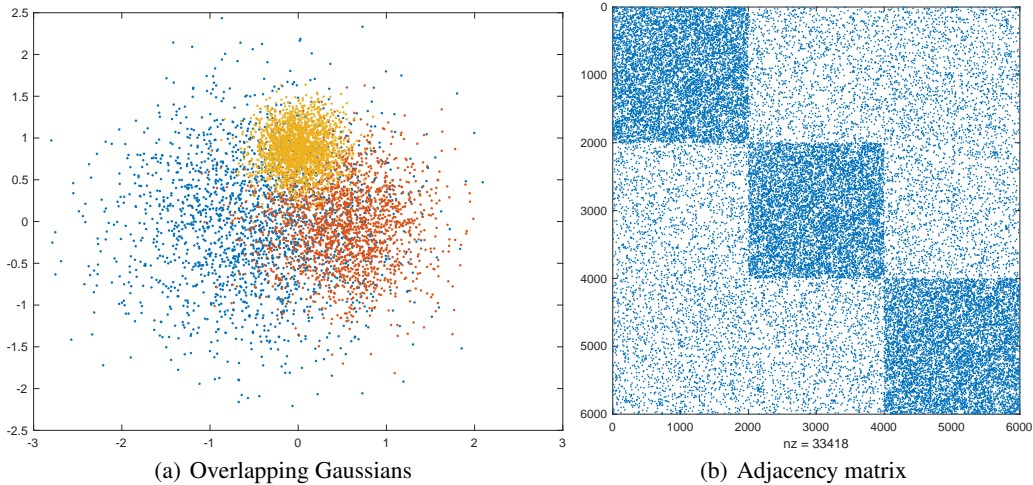

(a) Overlapping Gaussians          (b) Adjacency matrix

Figure 4: The "Mixture" data set (input features and graph).

### B.2  SUMMARY OF DATA SETS

See Table 2 for a summary of the data sets used in this work.

Table 2: Data sets.

|              | Mixture | Cora  | Pubmed |
|--------------|---------|-------|--------|
| # Nodes      | 6,000   | 2,708 | 19,717 |
| # Edges      | 16,709  | 5,429 | 44,338 |
| # Classes    | 3       | 7     | 3      |
| # Features   | 2       | 1,433 | 500    |
| # Training   | 2,400   | 1,208 | 18,217 |
| # Validation | 1,200   | 500   | 500    |
| # Test       | 2,400   | 1,000 | 1,000  |

Table 3: Hyperparameters for different GCN architectures and training algorithms.

(a) 1-layer GCN

|                   | Mixture | Cora  | Pubmed |
|-------------------|---------|-------|--------|
| Batch size        | 256     | 256   | 256    |
| Regularization    | 0       | 0     | 0      |
| SGD learning rate | 1e+0    | 1e+3  | 1e+3   |
| Adam learning rate| 1e-2    | 1e-1  | 1e-1   |

(b) 2-layer GCN

|                   | Mixture | Cora  | Pubmed |
|-------------------|---------|-------|--------|
| Batch size        | 256     | 256   | 256    |
| Regularization    | 0       | 0     | 0      |
| Hidden unit       | 16      | 16    | 16     |
| SGD learning rate | 1e+0    | 1e+2  | 1e+1   |
| Adam learning rate| 1e-2    | 1e-1  | 1e-1   |

## B.3 (HYPER)PARAMETERS

See Table 3 for the hyperparameters used in the experiments. For parameter initialization, we use the Glorot uniform initializer (Glorot & Bengio, 2010).

## B.4 RUN TIME

See Table 4 for the run time (per epoch). As expected, a smaller sample size is more computationally efficient. SGD with consistent gradients runs faster than the standard SGD and Adam, both of which admit approximately the same computational cost.

Table 4: Time per epoch in seconds.

|              | 1-layer GCN |        |        | 2-layer GCN |        |        |
|--------------|---------|--------|--------|---------|--------|--------|
|              | Mixture | Cora   | Pubmed | Mixture | Cora   | Pubmed |
| SGD (400)    | 0.0035  | 0.0269 | 0.1991 | 0.0103  | 0.0868 | 2.5014 |
| SGD (800)    | 0.0018  | 0.0455 | 0.3554 | 0.0103  | 0.0974 | 2.5684 |
| SGD (1600)   | 0.0027  | -      | 0.7129 | 0.0142  | -      | 3.2032 |
| SGD (3200)   | -       | -      | 1.1847 | -       | -      | 3.8895 |
| SGD unbiased | 0.0044  | 0.0737 | 2.2425 | 0.0130  | 0.2031 | 7.9478 |
| Adam unbiased| 0.0049  | 0.0741 | 2.2313 | 0.0143  | 0.2080 | 7.9037 |

