# OpenReview forum: "Stochastic Gradient Descent with Biased but Consistent Gradient Estimators"
_ICLR.cc/2020/Conference — Reject_

### Official Review · AnonReviewer2 · 2019-10-22
**Official Blind Review #2**

**Rating:** 1

**Review:**

The paper studies stochastic optimization with consistent (may not be unbiased) estimators. This problem is well-motivated through the example of learning graph representations where consistent estimators are easier to obtain than unbiased one. Under the assumption that the estimate converges to the consistent gradient exponentially fast w.r.t. the sample size, the authors give convergence rates for convex, strongly-convex and non-convex optimization. The authors validate their theory through synthetic experiments.

Overall, the paper is well-motivated and well-written however it lacks technically novelty. Under the assumption of exponentially fast convergence to small error, the setup is more like gradient descent (have access to approximate full gradient) than stochastic gradient descent as the paper supposes. The main convergence theorems seem to follow from standard techniques for inexact/noisy gradients. In [1], convergence rates for various first-order methods are proven under the assumption that the error is additive, that is, ||g - h|| <= \delta. Since the authors implicitly convert the multiplicative error to an additive error in their analysis, their assumptions are comparable to [1]. Also, since the analysis is more like GD, in the strongly-convex setting one can actually get faster convergence rates (logarithmic) as long as \delta is small (in comparison to the strong convexity parameter) unlike the O(1/T) ones mentioned in the paper.

Additional comments:
Assumption - There should be an additive error along with the multiplicative error as in the current setup. If ||h|| is very small then according to the assumption, the estimates of the gradient are very tight; this may not be true. Also, this assumption seems to only be needed for sample complexity purposes, making the tails weaker would only give a larger sample complexity without affecting the convergence rates. Would be good to separate these.

Convergence Analysis - As mentioned above, since the setting is like GD with noisy gradients, a more careful analysis in the strongly convex setting can improve the convergence result. Refer [2] for the standard analysis without noise.

Upper bound on l -  In Thm 2, the authors assume l <= G/||w_1 − w^*||. Why is this needed? Increasing l should make the problem more convex and easier.

[1] Devolder, Olivier, François Glineur, and Yurii Nesterov. "First-order methods of smooth convex optimization with inexact oracle." Mathematical Programming 146, no. 1-2 (2014): 37-75.
[2] Robert M. Gower. Convergence Theorems for Gradient Descent. https://perso.telecom-paristech.fr/rgower/pdf/M2_statistique_optimisation/grad_conv.pdf

**Experience Assessment:**

I have read many papers in this area.

**Review Assessment: Checking Correctness Of Derivations And Theory:**

I carefully checked the derivations and theory.

**Review Assessment: Checking Correctness Of Experiments:**

I assessed the sensibility of the experiments.

**Review Assessment: Thoroughness In Paper Reading:**

I read the paper at least twice and used my best judgement in assessing the paper.

---

> ### Author Response · Authors · 2019-11-12
> **Response to Official Blind Review #2**
>
> Thank you very much for the critical comments. We respectfully debate regarding the significance of the paper, as well as the connection to deterministic gradient descent and the gradient error assumption. We also answer other comments and update the paper with an additional result regarding linear convergence. We will be happy to discuss further to achieve a consensus understanding. Thank you.
>
> RE: Significance of the work.
>
> Technical matter aside, we consider that the primary contribution of the work is the introduction of a novel problem of practical significance in machine learning, together with a framework with which the problem may be approached. As outlined in Section 1.1, unbiased gradients are not always a viable choice in training machine learning models. Consistent gradients are one rescue, whose effectiveness has been corroborated in the training of graph convolutional networks (Chen et al., 2018). We formalize the problem in this paper and show that traditional analysis of stochastic gradients can be adapted when probability concentration is taken into account (as is often the case by using Chernoff bounds or existence of moments). We do believe that the problem of consistent gradients is a healthy introduction to the field, as evident in part by the increasing independent citations of the work prior to being formally published.
>
> RE: More similar to gradient descent than stochastic gradient descent?
>
> It is true that if the gradient error is sufficiently small, consistent gradient behaves like full gradient and hence the theory should be consistent with that of deterministic gradient descent asymptotically. However, such as a viewpoint does not devalue the theory of consistent gradient. Here is why. One may similarly argue that for unbiased gradient, if the variance is sufficiently small then it also behaves like full gradient. But this understanding should not undermine the significance of stochastic gradient descent theory developed over the past several decades. Drawing analogy, the minibatch size affects the variance of the unbiased gradient estimate; and the sample size affects the failure probability for the use of consistent gradient. It is these interesting factors that make stochastic gradient descent and our work distinct from deterministic gradient descent.
>
> RE: Faster convergence rate.
>
> Indeed one may prove linear convergence for the strongly convex case, at the cost of a positive right-hand side term that prevents the error from going to zero. This situation is similar to standard SGD, evidenced by Theorem 3.2 of the paper [2] that you reference, as well as other papers such as Bottou et al. (2016).
>
> Note that the value of such results is debatable. These results mean that in the limit, the difference between the iterate and the optimal solution is bounded by a nonnegligible positive term. However, in fact, both standard SGD analysis and our analysis with consistent gradients show that the difference can go to zero, albeit at a sublinear speed. In this regard, linear convergence bounds are much weaker than sublinear convergence bounds. On the other hand, one may argue from a machine learning perspective, that optimization error should be balanced with model class error and sampling error. In this perspective, having a nonzero optimization error is not that bad and the crucial point is that one may attain this error in a linear rate. We acknowledge both viewpoints.
>
> We have included the linear convergence result and discussions in the updated version of the paper. See Theorem 4.
>
> RE: Assumption on multiplicative gradient error.
>
> We believe that this assumption functions as well as the use of the additive form. Changing from the multiplicative form in (5) to an additive form only means that the gradient error is scaled by ||h_k||. Such a scaling can be absorbed in the right-hand side, leading to a different expression for the sample size N_k (that is, the new N_k is related to the old N_k by some expression of ||h_k|| and \delta). Such a change does not offer new insight but will make subsequent bounds look extremely complicated. Hence, using the additive form does not seem a good idea.
>
> You mention that “If ||h|| is very small then according to the assumption, the estimates of the gradient are very tight; this may not be true.” We agree that when ||h|| is small, the probability may be small given the same \delta, but this probability will be reflected in the sample size N_k, so *IT IS ALWAYS TRUE*.
>
> We also concur that some work uses the additive form, such as reference [1] you mentioned. However, specifically for that work, it analyzes a proximal-kind of algorithm rather than the standard (stochastic) gradient algorithm. Although both are first-order methods, the technique does not seem straightforwardly extensible from one to the other.
>
> Reference [1] is nevertheless valuable and we have cited it in the updated paper.

---

> > ### Comment · AnonReviewer2 · 2019-11-13
> > **Response to Authors**
> >
> > Thank you for addressing my comments. I still stand by my critiques.
> >
> > RE: Significance of the work. More similar to gradient descent than stochastic gradient descent?
> > I do not deny the value of consistent gradient estimators. Consistent but biased estimators have been used in previous literature [1]. The point I am raising is that under the assumption that your estimator error is small (Theorems assume \delta = 1/T), the case is not much different from GD (as you point out too). Usually, in SGD analysis, the variance is bounded by a constant not 1/T.
> >
> > RE: Faster convergence rate.
> > If you do the analysis more carefully, the second term should depend linearly with \delta and there should be no positive term in the limit. Intuitively, as \delta -> 0, the case is exactly gradient descent where you do get logarithmic convergence with no additional positive term.
> >
> > RE: Assumption on multiplicative gradient error.
> > I do not understand 'We agree that when ||h|| is small, the probability may be small given the same \delta, but this probability will be reflected in the sample size N_k, so *IT IS ALWAYS TRUE*.'
> > What I was trying to imply was that, for any constant \delta, observe that when your gradient is less than \eps/\delta, then your approximation is \eps close. However, \eps does not show up in your sample size. With just a constant samples, you cannot hope to achieve this for all values of h. It does not change the analysis significantly but is worth pointing out in my opinion.
> >
> > RE: Upper bound on l
> > I understand, thanks for making a note.
> >
> > [1] Daskalakis, Constantinos, Nishanth Dikkala, and Ioannis Panageas. "Regression from dependent observations." In Proceedings of the 51st Annual ACM SIGACT Symposium on Theory of Computing, pp. 881-889. ACM, 2019.

---

> > > ### Author Response · Authors · 2019-11-14
> > > **Response to response**
> > >
> > > Thank you for the follow up. We would like to debate and clarify each point for promoting understanding. We have also updated the paper by slightly patching (6) and (7).
> > >
> > > RE: Significance of the work. More similar to gradient descent than stochastic gradient descent?
> > >
> > > Consistent estimators are a standard subject in statistics; they have a long history. However, we believe that its use in optimization is novel. The new reference [1] you mentioned is concerned with whether the optimization result (at global optimum) is consistent with the ground truth model parameters, which is a completely different setting from ours that is concerned with how to approach a local/global optimum. In other words, [1] studies the consistency of the found w regardless of optimization, whereas we study the consistency of the gradient with respect to w.
> > >
> > > The shrinking \delta = \rho/T is a technical assumption. Essentially, \delta does not enter the bound; but rather, it is the sample size that controls the bound. For a desired failure probability, the sample size needs to be changed (see (7)) when considering the asymptotics with respect to T. Or equivalently, if one wants to keep a fixed sample size, then the failure probability deteriorates for larger T. The technical assumption is unique to the nature of consistent estimators, whose analysis is an interplay of estimator error, sample size, and failure probability.
> > >
> > > RE: Faster convergence rate.
> > >
> > > The intuition that as \delta -> 0, the case is exactly gradient descent is challenging to be translated into a theoretical result. As a defense we would use unbiased SGD as an example; similar things have not been seen there yet. Specifically, for unbiased SGD one would imagine that as the gradient variance -> 0, it is exactly gradient descent. But not yet. Consider the past reference [2] you mentioned https://perso.telecom-paristech.fr/rgower/pdf/M2_statistique_optimisation/grad_conv.pdf . See Theorem 3.2. The right-hand side has a nonzero term related to B. Because B is the expectation of the gradient norm, and because the gradient norm is nonnegative, letting B be zero requires that the gradient is zero in all optimization steps, which is unrealistic.
> > >
> > > We are open minded and welcome to see linear convergence results with a vanishing right-hand side. But before we see one for unbiased SGD, it is perhaps asking too much for consistent SGD.
> > >
> > > RE: Assumption on multiplicative gradient error.
> > >
> > > Thank you for clarifying that it is unrealistic to assume that the sample size admits the same bound for each step. We agree and we patched the assumption (see (6) and (7)). One does not need to change from a multiplicative form to an additive form, which makes subsequent bounds overly complicated. Rather, we consider that the coefficient C needs be step-dependent (and hence we modified it to C_k). The essence is that for a given estimator error and sample size, the failure probability should vary for different steps. This variation is reflected by the coefficient C_k.

---

> > > > ### Comment · AnonReviewer2 · 2019-11-14
> > > > **Response**
> > > >
> > > > Thank you for your response.
> > > >
> > > > "The shrinking \delta = \rho/T is a technical assumption".
> > > > This is a strong assumption since you assume \rho is constant. Hence if you want good convergence guarantees you need T to be large and therefor \delta to be pretty small. This implies you need large sample complexity. This is why I believe the setting you are analyzing is pretty close to GD. Reviewer 3 agrees with this point.
> > > >
> > > > "Faster convergence rate"
> > > > I am confident it is possible for the strongly convex case. The unbiased gradient analysis in [2] does not use the knowledge of closeness (existence of \delta) that your analysis explicitly uses.
> > > >
> > > > I agree the addition change makes it complicated but it is worth pointing out that it is necessary for small values of ||h||. I am not sure how C_k handles that since it still requires the multiplicative property to hold.

---

> > > > > ### Author Response · Authors · 2019-11-15
> > > > > **Response to response**
> > > > >
> > > > > Thank you. We have updated the paper with a new subsection 4.4 and Theorem 7, which should address your latest comments simultaneously.
> > > > >
> > > > > Theorem 7 indicates that indeed for consistent SGD, it is possible to obtain linear convergence rate with a diminishing right-hand side. This result is based on the work of Friedlander & Schmidt mentioned by Reviewer 3 in a recent post. In this result, if \delta goes to zero, then the right-hand side bound goes to zero, well agreeing with the deterministic GD case. It still uses the assumption (6). The multiplicative property is translated to a term \delta_k/||h_k|| in the sample size bound. The use of this term depends on the relative decrease speed between \delta_k and ||h_k||.
> > > > >
> > > > > To obtain diminishing right-hand side, we need to make additionally a smoothness assumption. This assumption is crucial (at least in the work of Friedlander & Schmidt).
> > > > >
> > > > > One point we would like to stress is that if a stronger SGD result is brought to our awareness later (such as not requiring smoothness), we could probably adapt that result to the consistent gradient case. However, this is an endless game. Chasing the best convergence result and weakest assumption destroys the purpose of the paper. This is the first work that studies consistent gradients. Our aim is to introduce the problem to the field and engage research. Our message is that for this new setting the convergence rates can be maintained the same as existing settings (deterministic GD, unbiased SGD). There are practical applications (graph neural networks, and learning to rank) behind it. This work inspires the development of new training algorithms (such as the case Reviewer 1 brought up; see the end of the conclusion section). Furthermore, we present results that are correct and citable (and researchers did cite). For these reasons, we would argue that the novelty and contribution lie in the problem and the results more than in the proof technique.

---

> ### Author Response · Authors · 2019-11-12
> **Response to Official Blind Review #2 (continued)**
>
> RE: Upper bound on l.
>
> The purpose of having an assumption on l is only to make the right-hand side of the bound look slightly cleaner. This assumption is not crucial. Without this assumption, the right-hand side will be the max of two terms. One is the current term, and the other is related to G^2/l^2, because ||w_1 – w^*|| is the base case of an inductive proof. Making l larger will decrease G^2/l^2, rendering a diminishing role in the presence of a max.
>
> We also note that l can be made arbitrarily small. That is, if f is l-strongly convex, it is also l’-strongly convex, for all l’ < l. Hence, the assumption is quite mild.
>
> We have inserted a short discussion regarding l after Theorem 2.

---

### Official Review · AnonReviewer1 · 2019-10-25
**Official Blind Review #1**

**Rating:** 6

**Review:**

The authors study SGD algorithms for problems where obtaining unbiased gradients is potentially computationally expensive. In such cases while obtaining, unbiased gradients is expensive, it might be possible to establish consistent estimators of the gradient. The authors then establish that SGD algorithm when run with consistent gradient estimators (but not necessarily unbiased) have similar convergence properties as SGD algorithms when run with unbiased gradient estimators.  The example problem class considered is the problem of learning embeddings for graph problems, where the task is to get embeddings for nodes. Such embeddings can be used to do node classification or solve any other downstream task that involves the nodes of the graph. For such graph problems learning embeddings requires us  to look at the neighbours of a node, neighbours-of-neighbours and so on, which means that in the worst case calculating gradient w.r.t. a single node can be of time complexity O(N).  Consistent gradient estimators have been proposed for such graph problems in the past but this paper establishes theoretical properties of SGD with such estimators.

The paper is well written and the results are convincing. I have a few questions/comments

1. In all the experimental results the loss curves are shown w.r.t. the number of epochs. It is clear that using unbiased SGD, unbiased ADAM is better of than using biased SGD. However, these plots do not tell the complete story as the key point behind using consistent SGD is not achieving lower loss, but actually faster computation. I would suggest that the authors show run-time plots that show how the run-time scales with epochs.

2.  I appreciate the authors efforts in explaining their assumptions and how different assumptions kick in.

3. I wonder if a similar methodology can be applied even to the case of ranking problems (say rank net, see reference below). In ranknet training proceeds via  choosing a pair-of-documents and performing gradient updates w.r.t. the pair. However, if one were to pick a single document, the gradient update w.r.t. that document (d1) should involve all other documents (d2) that are less relevant than d1. My question is does applying the consistent gradient methodology in this paper reveal a new algorithm for training ranknets?

**Experience Assessment:**

I have read many papers in this area.

**Review Assessment: Checking Correctness Of Derivations And Theory:**

I assessed the sensibility of the derivations and theory.

**Review Assessment: Checking Correctness Of Experiments:**

I assessed the sensibility of the experiments.

**Review Assessment: Thoroughness In Paper Reading:**

I read the paper at least twice and used my best judgement in assessing the paper.

---

> ### Author Response · Authors · 2019-11-12
> **Response to Official Blind Review #1**
>
> Thank you very much for the suggestive comments. We appreciate them a lot.
>
> RE: Time versus epoch.
>
> Indeed, the advantage of consistent SGD lies in the smaller computational cost. We do, however, caution that (1) a lower loss does not necessarily mean a better classification accuracy, in light of the balance between optimization error, hypothesis class error, and sampling error; and (2) in practice, early stopping is used when one observes that the validation error (not training error) does not improve. Hence, timing is a tricky game. An extensive timing comparison has been reported by FastGCN (Chen et al. 2018). In this paper, for completeness we report the per epoch time to validate the computational advantage of sampling. This new information is in Appendix B.4 of the updated paper.
>
> RE: Application to ranking.
>
> Very interesting application and the answer is affirmative! Consider eqn (8) of the paper https://papers.nips.cc/paper/2971-learning-to-rank-with-nonsmooth-cost-functions.pdf . Here,  s_i means the ranking function of document i (and similarly for s_j) and C_{i,j}^R is the ranking loss for a pair {i,j}. The overall RankNet loss is a double sum over i and j. The referenced paper proposes speeding up training by taking the sum over j first and then over i (see eqn (7)). This process is still O(n^2) but the authors argue that the n^2 calculation of C_{i,j}^R is far cheaper than the O(n) forward computation of s_i and O(n) backward propagation. By using our methodology, we propose a stochastic training whereby one samples i and j separately and backpropagates the mini-batch sample loss. Because of nonlinearity in (8), the separate samplings of i and j do not result in an unbiased gradient; rather, the gradient estimate can be consistent, following a similar argument to the proof of Theorem 1 in our paper. Then, our stochastic training theory applies.
>
> We have included such discussion in the conclusion section of the paper.

---

> > ### Comment · AnonReviewer1 · 2019-11-15
> > **Satisfied with the rebuttal**
> >
> > I am satisfied with the rebuttal and appreciate the authors efforts to add new plots and the new application,

---

### Official Review · AnonReviewer3 · 2019-10-30
**Official Blind Review #3**

**Rating:** 3

**Review:**

The analysis of the convergence of SGD with biases gradient estimates dates back to Robins&Monroe, but the authors of this paper focused on a recent original algorithm that shows that once can estimate the approximate gradient of a large GNN network, simply by sampling nodes randomly.

When I first read of the paper, I was enthusiastic because I did not know the FastGCN approach presented at ICLR the previous year, which showed that the gradient of a GCN could be efficiently approximated by sampling a subset of the nodes. After reading FastGCN, I was less enthusiastic as most of the originality relied on the consistent estimate of the gradient, when t (number of sampled in the neighbours of the output nodes) increases.

The main contribution of the paper is the proof that the algorithm converge, but there is no theoretical analysis of the key quantity "t", which is the number of sampled nodes in the neighbours of the output nodes. I would expect to see the number of sample grow as the algorithm converge to the optimal solution since the gradient needs less bias when the algorithm converges. However, the authors do not address this point.

I did not into the details of the proofs, but it seems to me that they are quite loose and several details such as the functional spaces, and the boundedness assumptions, are not mentioned. Here are few examples:
- In the first sentence: P(x, y) of data x and associated label y. The space of x, the space of y and the probability space are not defined. In fact, no set in which variables belong is defined in the paper.
- The Theorem 1 is strange to me. I would assume that one needs some assumption of boundedness of Q and finite moments for P to avoid pathological examples where the integral (for the asymptotic expectation) is infinite, but the finite sum G_{st} is always finite, contradicting the limit in theorem 1.

Overall, while proving that the FastGCN algorithm is consistent is important, it is hard to understand how useful the results are and how they can be useful in practice. For example, what can we interpret or what can we learn from the bounds given by Theorem 2?

Finally, I might miss something, but the empirical results showed do not seem to show better gains than the Adam algorithm. The theory shows that the more bias we have, the less accurate we should be, why isn't it apparent in Table 1. Is there something such as the computational cost of Adam, that I'm missing, especially when looking at the graphs? I'm sorry if I did not get the main message of the experiments, but even after reading the paper 3 times, I did not understand what the authors wanted the reader to conclude with these experiments.


**Experience Assessment:**

I have published one or two papers in this area.

**Review Assessment: Checking Correctness Of Derivations And Theory:**

I assessed the sensibility of the derivations and theory.

**Review Assessment: Checking Correctness Of Experiments:**

I assessed the sensibility of the experiments.

**Review Assessment: Thoroughness In Paper Reading:**

I read the paper at least twice and used my best judgement in assessing the paper.

---

> ### Author Response · Authors · 2019-11-12
> **Response to Official Blind Review #3**
>
> Thank you very much for the comments. In what follows, we address them point by point. We will be happy to clear your doubts should more questions arise.
>
> RE: Biased, unbiased, and consistent.
>
> At the beginning of your comments, you mentioned that analysis of SGD with biased gradient dates back to Robins & Monroe. It may be a typo, but in case confusion arises, we would like to stress that along the history, most analysis was for the *unbiased* case, rather than the biased case. The biased case was sporadically addressed while analysis of the consistent case was never seen before. The novelty and main contribution of this paper is the analysis of the consistent case.
>
> A minor note is that, whereas we prove that in FastGCN the gradient estimator is consistent, it is only one motivating application; indeed, Reviewer #1 even suggested another application, training RankNets, which we have added to the discussion in the paper. Our focus is to prove optimization convergence under consistent gradient estimators, regardless what application they are meant for.
>
> RE: Sample size “t” and convergence, part 1.
>
> Sample size is indeed a key quantity. First, we note that the sample size does not need to increase as the algorithm approaches convergence. The gradient does not need to be less and less biased, either. The analogy to the standard unbiased SGD is that the gradient variance does not need to shrink on approaching convergence. For our case, the analysis is based on convergence under high probability, as elaborated below.
>
> RE: Sample size “t” and convergence, part 2.
>
> The analysis strategy we employ is to show that with probability related to the sample size, convergence occurs at a certain rate. To conform to notational convention, in most of the places we use N to denote the sample size. In all theorems, the probability (1-\epsilon) under which convergence occurs is related to N in equation (6). Just like most of the concentration analysis in the literature, the results read that with a sufficiently large sample size N, with probability at least 1 - epsilon(N), a certain fact holds. Our results follow the same format.
>
> RE: Space of P(x,y).
>
> The notion of joint probability distribution P(x,y) of data x and label y comes from standard statistical learning theory. One may assume that x belongs to a vector space (e.g., R^d) and y is a real-valued scalar either bounded or unbounded, and either continuous or discrete. P is a probability distribution over the product space of x and y. In reality, P is unknown.
>
> RE: Assumptions of Theorem 1.
>
> The most important assumption for Theorem 1 to hold is that q is continuous (as stated), since the proof invokes the continuous mapping theorem. The proof also invokes the law of large numbers, which could either be the weak law or the strong law, since the plim result follows the weak law. Both laws require only that the expectation is finite (eqn (4)). No higher order moments need be finite and no boundedness assumption need be made.
>
> We have updated the paper with the assumption that f is finite.
>
> RE: Interpretations of results (e.g., Theorem 2).
>
> Overall, our convergence results provide a guarantee that running SGD with consistent gradient estimators will converge, just like the traditional case of unbiased gradient. But unlike the traditional case, convergence does not occur in the expectation sense; rather, it occurs in a probabilistic sense. The more samples one uses to estimate the gradient, the higher the probability that convergence occurs. Meanwhile, the convergence rate is the same as for the case of unbiased gradients.
>
> For example, Theorem 2 states that the convergence rate is O(1/T) and the probability is 1-\epsilon, where \epsilon is a decreasing function of the sample size N in equation (6).
>
> RE: Empirical comparison with Adam and computational cost.
>
> The purpose of this paper is not to propose an algorithm faster than Adam, but rather to show that if one is unable to efficiently compute the unbiased gradient estimates that Adam needs, then it is still possible to use other gradient estimators such that SGD converges. In the FastGCN case, such a gradient estimate comes from sampling the neighbors and it is a consistent estimator. Indeed, we do not expect that SGD converges to a better place than does Adam. The point is that computing the consistent gradient estimate is more efficient than computing the unbiased estimate, as elaborated in Sections 1 and 2. Further, please see timing results in Appendix B.4 that validate the computational advantage.
>
> RE: Table 1.
>
> As such, using consistent gradient estimators will not result in more accurate parameters or classification accuracies. They are just faster to compute. The numbers in Table 1 show that while one uses less effort to perform the computation, the accuracies are not sacrificed.

---

> > ### Comment · AnonReviewer3 · 2019-11-14
> > **Useful responses, but paper of limited impact**
> >
> > I thank the authors for the details response.
> >
> > The authors reminded in their response to my review that their goal is "to show that it still possible to use other gradient estimators such that SGD". I understand, but again, I think there is nothing fundamentally new and more work would be required to clearly understand and control the bias and variance of the proposed biased gradient estimate (e.g. I expect that the more "non-linear" the GNN transformation are, the higher the bias of the gradient estimate).
> >
> > About history: I agree the original Robbins&Monro paper from 1951 did not include bias term, but people quickly realised that unbiased gradient estimates where hard to obtain (e.g. in control theory) and the theory was quickly adapted to biased gradient, the most well-known one being Kiefer-Wolfovietz algorithm in 1952. I would strongly recommend the authors to review this algorithm and its subsequent extension, such as the Spall method, to estimate a biased but consistent gradient. Note also that most of the "traditional" courses on stochastic optimisation are based on the general theorem from Dvoretzky's in 1956 that includes biases in the gradient and more exactly the bound on the "Transformation variable" called "T_n" in the paper https://projecteuclid.org/download/pdf_1/euclid.bsmsp/1200501645

---

> > > ### Author Response · Authors · 2019-11-14
> > > **Further clarification**
> > >
> > > Thank you for confirming the technical correctness of the work. The impact of the paper is a subjective matter. We have made clear the significance of the paper and we respect your opinion.
> > >
> > > We would like to address two points you raised above to promote understanding.
> > >
> > > RE: Bias and variance.
> > >
> > > The bias-variance tradeoff is a concept about models but consistency is a concept about random variables. These two concepts are unrelated and do not seem to be applicable to each other’s scenario. Consistency is about the convergence of a sequence of random variables to a limit (if it exists) and how fast the convergence is. Increasing the sample size is one way to construct this sequence. From Figure 3, the failure probability decreases faster when the neural network has fewer layers. In a sense, the more layers, the harder the problem (meaning needing more samples to maintain the failure/success probability).
> > >
> > > Additionally, we would like to point out that the control of bias-variance tradeoff, even if it exists, is an algorithmic issue, which is orthogonal to the aim of this paper---analysis of a given algorithm.
> > >
> > > RE: Kiefer--Wolfowitz.
> > >
> > > The essence of Kiefer--Wolfowitz is to replace the unbiased gradient estimate by its first-order finite difference. The finite difference incurs a bias and Kiefer--Wolfowitz requires this bias to vanish, together with other conditions, to ensure convergence. In fact, the paper you mentioned proves almost sure convergence and convergence in mean, without rates, but we, as argued next that the setting is rather different, prove convergence in probability with convergence rates. From almost sure convergence and convergence to mean to convergence in probability is straightforward, but giving rates is more interesting.
> > >
> > > Now, about the difference between Kiefer--Wolfowitz (and subsequently Spall and related methods) and our work. Excusing tedious mathematical notations, let us say the gradient of f is approximated by [f(x+c)-f(x-c)]/(2c). To prove convergence, Kiefer--Wolfowitz assumes that c vanishes eventually (thus the bias vanishes). For us, the notation of biasedness is not relevant, because as discussed at the very beginning (section 1.2), even when an estimator is consistent, it could still be asymptotically biased. The amenable mathematical tool is to exploit the diminishing probability when the estimator error decreases. Under this context, all our convergence results happen under a certain probability. This probability depends on the use of sample size to control how far apart the estimator is from the true gradient.

---

> > > > ### Comment · AnonReviewer3 · 2019-11-14
> > > > **Udpate**
> > > >
> > > > Thanks for your answer to my review feedback.
> > > >
> > > > "the more layers, the harder the problem (meaning needing more samples to maintain the failure/success probability)"
> > > >
> > > > I would love to see a link between the number of layers and the failure/success probability, but the results in the paper do not show that as there is no variable related to the number of layers in the theoretical results. The authors mention "theoretical results for more layers straightforwardly follow that of Theorem 1 below, through induction.", but this is not shown and does not look that straightforward to me....
> > > >
> > > > "This probability depends on the use of sample size to control how far apart the estimator is from the true gradient."
> > > >
> > > > I read again the paper and the sample size does not increase with the number of iteration, so I do not see why this claim holds...
> > > >
> > > > I also read the review of AnonReviewer2 and it seems that his/her concerns mirror mines when discussing the statement "When \delta -> 0, the case should become gradient descent". I read that the authors agree that this is a challenging analysis, but nevertheless important in this context. Note that growing sample sizes to optimise convergence rate have been analysed in the past, and I guess it could easily be extended to the "biased gradient case", e.g. using the Friedlander&Schmidt approach:
> > > > Friedlander, Michael P., and Mark Schmidt. "Hybrid deterministic-stochastic methods for data fitting." SIAM Journal on Scientific Computing 34.3 (2012): A1380-A1405.

---

> > > > > ### Author Response · Authors · 2019-11-15
> > > > > **Further clarification and updates**
> > > > >
> > > > > Thank you for the updated comments. We welcome questions and we think they cultivate healthy communications for enriching knowledge and understanding.
> > > > >
> > > > > Overall, it appears that you get stuck on the assumption (6), for which we think it is worthwhile to stress a few points. (i) This inequality is for one step. To control an overall failure probability \epsilon with T steps, we want the right-hand side of (6) to be <= \epsilon/T. (ii) Then, simple algebraic manipulation leads to (7). What (7) means is that if one wants to use more steps T, then assuming the gradient error level \delta does not change, the sample size N_k indeed needs to be larger. (iii) In our theorems, \delta changes, however. Because of the unidentified function \tau, it is unclear how the sample size bound (7) will vary. If \tau is an increasing function and \delta is 1/T, then the bound increases as T increases. (iv) The cited statement regarding “\delta -> 0” is a different matter than the sample size bound. The question there is whether one can obtain linear convergence to zero for the strongly convex case. Your cited paper of Friedlander & Schmidt inspires us of this possibility, with additionally a smoothness assumption. We have updated the paper to include this result (see section 4.4), by using the assumption (6) but with a slightly different assumption on \delta from those of the theorems in section 4.2.
> > > > >
> > > > > In what follows we answer your other two questions.
> > > > >
> > > > > RE: link between the number of layers and the failure/success probability.
> > > > >
> > > > > What we stated earlier was an empirical observation. Asymptotically, the failure probability is smaller in the 1-layer GCN case compared with the 2-layer GCN case, for the same gradient error level \delta and same sample size N, since the solid curves bend over while the dashed curves are straight in Figure 3. This observation translates to the assumption (6) that the coefficient C_k is smaller for 1-layer GCN. Obtaining an expression of C_k with respect to the number of neural network layers is far beyond the scope of this work.
> > > > >
> > > > > RE: Induction proof of FastGCN.
> > > > >
> > > > > The proof that FastGCN yields a consistent gradient estimator is the subject of Chen et al. (2018). In the current work, we use FastGCN as a motivating application for the analysis of consistent SGD. The analysis is independent of the application (at the end of the paper we further give another application).
> > > > >
> > > > > If one wants to understand FastGCN, the induction proof is in fact not hard. Intuitively, for eqn (4), the integral that computes f and the integral that computes Z(v) for each v are Monte Carlo approximated. In between there is a nonlinear (but continuous) activation function \sigma(). When one sees Monte Carlo, one invokes the law of large numbers. When one sees an activation function, one invokes the continuous mapping theorem. If the network has several layers, then Monte Carlo approximations and activation functions are interlaced. It is the repeated appearance of Monte Carlo and activation that needs induction.
> > > > >
> > > > > Again, FastGCN is not the contribution of this work; we recommend the readers to see Chen et al. (2018) for details.

---

### Official Review · AnonReviewer4 · 2019-11-01
**Official Blind Review #4**

**Rating:** 3

**Review:**

This paper aims to solve the stochastic optimization problems in machine learning where the unbiased gradient estimator is expensive to compute, and instead use a consistent gradient estimator. The main contributions are the convergence analyses of the consistent gradient estimator for different objectives (i.e., convex, strongly convex, and non-convex). Overall, it is interesting and important, but I still have some concerns.

Which objective function do the authors aim to minimize, the expected risk or empirical risk? I guess it's the empirical risk, right? If so, the sample size $N$ is constant. This may break the condition (sufficiently large sample sizes satisfying (6)) of theorems (i.e., Theorem 2, 3, 4, 5). Thus, it will narrow the application domains of the theorem.

For the proofs of theorems, the main difference between SGD and this paper is the involvement of Lemma 9, which is one part of the assumption. Besides, this assumption involves the failure probability $\epsilon$. The convergence theorem (e.g., Theorem 2) has a probability condition to hold the Eq.(7) (or (8)). Maybe some comments below the theorem can be discussed to decrease $\epsilon$, although authors discuss it in experiments ("This phenomenon qualitatively agrees with the theoretical results; namely, larger sample size improves the error bound.").

For the experiments, the authors focus on the training of GCN model. I think it can be considered a doubly stochastic sampling process, which is one for the sample and the other for its neighbor. Is that right? Besides, for Figure 1, can the "SGD unbiased" be viewed as "SGD consistent (sampl $n$)"? If not, I think it's important to compare these two because this will clearly show the performance difference between the unbiased and consistent estimator.


**Experience Assessment:**

I do not know much about this area.

**Review Assessment: Checking Correctness Of Derivations And Theory:**

I assessed the sensibility of the derivations and theory.

**Review Assessment: Checking Correctness Of Experiments:**

I assessed the sensibility of the experiments.

**Review Assessment: Thoroughness In Paper Reading:**

I read the paper at least twice and used my best judgement in assessing the paper.

---

> ### Author Response · Authors · 2019-11-12
> **Response to Official Blind Review #4**
>
> Thank you very much for the questions. In what follows we answer them one by one and hope that the replies help reassess the work.
>
> RE: Expected risk or empirical risk?
>
> Regarding the sample size, indeed the framework of consistent gradient is more amenable to the expected risk setting, because the sample size in empirical risk is often finite. However, this is not always the case. One may consider an online setting where samples are streamed in. For example, for dynamic graphs where the node set endlessly grows, one may face a large graph such that under a desired probability, the sufficiently large sample size is met. In practice, the required sample size is often overly conservatively estimated. As we have shown in the numerical examples, even a small sample size yields sufficiently good classification results. Furthermore, even when the sample size is finite, the bounds are still useful and can be used to say with what probability we expect convergence to hold.
>
> RE: Decreasing the probability \epsilon.
>
> Indeed, like most work in other fields based on probability concentration (e.g., in the field of randomized linear algebra), the failure probability \epsilon is a quantity controlled by the sample size. Such discussion also appears in Section 4.3 following the theorems.
>
> RE: Doubly stochastic process.
>
> In a sense, the training of the GCN model in the FastGCN way may be interpreted as a doubly stochastic process, bearing certain similarity with the Doubly Stochastic Gradients work by Dai et al in NIPS 2014. In their work, one stochasticity occurs in the sample and the other occurs in the approximation of the kernel. A major distinction is that in their work, the double stochastic sampling remains unbiased because the functional gradient is linear, whereas in our case, because of the nonlinearity of the neural network, unbiasedness is lost. One obtains only consistency. In fact, the lack of unbiasedness is the major motivation for us to develop theory for consistent gradients.
>
> In the particular case of GCN, indeed “SGD unbiased” implies “SGD consistent” (note, no sample size regarding neighbors is involved in the unbiased estimator). One may understand this fact by simply invoking the law of large numbers on the samples.

---

### Author Response · Authors · 2019-11-12
**Summary of updates in the paper**

- We patched an assumption to Theorem 1.
- We inserted a discussion of an assumption of Theorem 2. See text afterward.
- We patched (6) and (7).
- We added a linear convergence result (and proof) for the strongly convex case. See Theorem 4.
- We added a remark section 4.4 that discusses linear convergence result.
- We inserted a new application (ranking) in the conclusion section.
- We added two references regarding noisy gradients.
- We included timing results in Appendix B.4.

---

### Decision · Program_Chairs · 2019-12-19

**Decision:**

Reject

**Comment:**

This paper analyzes the convergence of SGD with a biased yet consistent gradient estimator. The main result is that this biased estimator results in the same convergence rate as does using unbiased ones. The main application is on learning representations on graphs (e.g., GCNs), and FastGCN is a closely related work. I agree that this paper has valuable contributions, but it can be further strengthened by considering the review comments, such as on the key assumptions.